# Circannual Clock in *Laelia speciosa* (Orchidaceae) Through Dormancy vs. Germination Dynamics of Seeds Stored Under Controlled Conditions

**DOI:** 10.3390/plants14030336

**Published:** 2025-01-23

**Authors:** Erandeni Durán-Mendoza, Martha Cornejo-Gallegos, Alejandro Martínez-Palacios, Martha Elena Pedraza-Santos, Nahum M. Sánchez-Vargas, Guadalupe Alejandra Valdovinos-Ramírez, Adelaida Stephany Hernández-Valencia, Juan Manuel Chavarrieta-Yáñez, Eloísa Vidal-Lezama, María del Carmen Mandujano-Sánchez

**Affiliations:** 1Technological Institute of the Valley of Morelia of the Technological Institute of Mexico, Morelia 58120, Mexico; erandeni.dm@morelia.tecnm.mx; 2State Commission for the Development of Indigenous Peoples of the Government of the State of Michoacán, Guadalajara 44340, Mexico; mcor78gallegos@gmail.com; 3Institute of Agricultural, Livestock and Forestry Research, Michoacana University of San Nicolas de Hidalgo (IIAF-UMSNH), Morelia 58000, Mexico; nahum.sanchez@umich.mx (N.M.S.-V.); 1341068b@umich.mx (G.A.V.-R.); manuel.chavarrieta@umich.mx (J.M.C.-Y.); 4Faculty of Agrobiology, Michoacana University of San Nicolas de Hidalgo (UMSNH), Morelia 58000, Mexico; martha.elena.pedraza@umich.mx; 5Plant Pathology at the Institute of Plant Health at the College of Postgraduates Campus Montecillo, Texcoco 56230, Mexico; hernandez.adelaida@colpos.mx; 6Department of Plant Science, Autonomous University Chapingo, Texcoco 56230, Mexico; evidall@chapingo.mx; 7Institute of Ecology, National Autonomous University of Mexico, Mexico City 04510, Mexico; mcmandujano@iecologia.unam.mx

**Keywords:** conservation, dormancy dynamics, in vitro, circannual biological clock

## Abstract

This study aimed to determine the dynamics of dormancy using triphenyl tetrazolium chloride (TTC) and asymbiotic germination in *Laelia speciosa* (Kunth) Schltr. seeds stored for three years and one year at different temperatures. This is the first report of a circannual rhythm in *L. speciosa* seeds under controlled storage conditions. Two experiments were carried out: (a) with seeds from wild populations of *L. speciosa* collected at two different times and dehydrated to 4% relative humidity (RH) and stored for three years at 25 °C, 6 °C, −20 °C, and −80 °C, and (b) with seeds from six fruits stored in liquid nitrogen (LN2; −196 °C) at 25 °C for 12 months. The germination conditions were 25 ± 1 °C with 16 h of light (23 μmol m^−2^ s^−1^) and 8 h of darkness for both trials. Because they have a rudimentary embryo, orchids are attributed a morphological latency; however, the staining of the embryo with TTC (>92%) in all the evaluations carried out throughout a year and the decrease in asymbiotic germination in the intermediate evaluations record a circannual biological cycle or clock, under temperature and humidity control (4% RH).

## 1. Introduction

*Laelia speciosa* (H. B. Kunth) Schltr. (Orchidaceae), regionally known as “*flor de mayo*” or “flor de Corpus Christi” due to Catholic religious festivities, is endemic to Mexico. It is distributed in the Sierra Madre Oriental and Sierra Madre Occidental, in the states of Michoacán, Guanajuato, Jalisco, Durango, Zacatecas, Aguascalientes, Querétaro, Hidalgo, San Luis Potosí, and Tamaulipas. This species blooms in May and June, with fruits taking approximately one year to mature and releasing seeds between April and May [1,2,3]. It is an epiphyte, primarily associated with *Quercus desertícola* Trel. (Fagaceae), and grows at altitudes of 1700 to 2300 m [1]. *L. speciosa* is listed under special protection (Pr) in NOM-059-ECOL-2010 [4] and in Appendix II of CITES [5]. The primary threats to its survival include deforestation, land-use changes, and the local collection of flowering plants for religious celebrations, as well as their sale to the public in urban areas by community members who harvest the plants during their blooming period [2]. The best conservation strategy is always in situ conservation [6]. However, when this is no longer feasible or to minimize future disturbance, ex situ seed banks are a viable alternative [7].

Advances in the conservation of orchid seeds in cold storage highlight the need to reduce relative humidity (<8%). This reduction enables seeds to withstand storage temperatures as low as −196 °C. However, most of the reports are to show germination at different temperatures [8,9]; at best, they report germination and dormancy behavior over a year [10]. Typical storage conditions involve low relative humidity at 6 °C, with viability reported for up to one or two decades. These results are not particularly encouraging, suggesting the need to explore lower temperatures that could extend seed viability [11,12].

This short viability (categorized as short-lived orthodox seeds) may be attributed to the fact that orchid seeds generally lack an endosperm and possess an undifferentiated embryo [13]. The cells of freshly mature seeds contain lipids and proteins but no starch [14]. In the best-case scenario, they may exhibit a rudimentary cotyledon [15]. As a result of evolution, orchid seeds sacrificed their food reserves and significantly reduced their size. Today, they are characterized by being extremely small, measuring 1–2 mm in length and 0.5–1 mm in width [16], with some species producing even smaller seeds [17]. Having tiny seeds could appear to be a negative feature, but the small seed size was compensated by evolving in association with mycorrhizal fungal symbionts [18], a strategy that has made them highly successful. Orchids are now considered cosmopolitan, thriving in virtually all environments where other plant groups can survive on the planet [19]. Orchid seeds have an undifferentiated embryo and show morphological dormancy or morphophysiological dormancy [20].

A dormant seed (or other germination unit) is one that lacks the ability to germinate within a specific timeframe under any combination of normal physical environmental factors (e.g., temperature, light/darkness) that would otherwise be favorable for germination [20]. Seed dormancy is an innate property that determines the environmental conditions under which seeds can germinate, determined by their genetics and closely influenced by environmental factors [21]. Dormancy is hierarchically reclassified according to [22] into class, level, and type. The system includes five classes of dormancy: physiological dormancy, morphological dormancy, morphophysiological dormancy, physical dormancy, and combinational dormancy (physical + physiological dormancy). Each class is subdivided into levels, and each level has different types. Many seeds exhibiting primary dormancy differences are considered to have morphological dormancy, which may be accompanied by physiological dormancy, a combination referred to as morphophysiological dormancy [23]. The complexity of dormancy continues to amaze, revealing the remarkable diversity of dormancy types in the plant kingdom [24].

Over the last three decades, breakthroughs in molecular genetics have uncovered the molecular mechanisms of a cellular circadian clock, rivaling the elegance and innovation of the remarkable timepieces crafted in the 18th century [25].

The circadian clock is an endogenous timekeeper present across all domains of life, enabling organisms to anticipate predictable daily changes in their surroundings [26,27]. However, the complexity of the clock and the diversity of its phenotypic outcomes indicate that it also plays an essential role, on a larger scale, in adapting to the seasonal variability of environmental conditions [28]. A period is defined as the time required to complete one cycle, typically measured from peak to peak [25]. In plants with circannual dormancy, this corresponds to an annual cycle [29]. The variability of the genetic clock can be considerably large between cultivars and their wild progenitors [30]. While the former may no longer respond to environmental changes, the latter retain these traits as natural survival strategies.

All taxonomic groups have developed genetically programmed timekeeping mechanisms. These regulate seasonal cycles in physiology and behavior that optimize survival and reproductive success and are formally described as circannual rhythms [29].

The basis of the biological clock (circadian and circannual) is based on studies of the genomics of organisms. Genes have been determined that are responsible for adjusting the chronology of the life cycle, expressing themselves in physiological, morphological, and behavioral changes in correlation with seasonal changes, as part of the evolutionary strategy favoring survival, with this expression being very marked in temperate zone environments [31,32,33].

In biology, circannual rhythms (from Latin *circa*, meaning ‘around’ and *annus*, meaning ‘year’) are oscillations of biological variables (physiological, morphological, and behavioral) at regular intervals of approximately 365.25 days, as survival strategies [33]. In this regard, although there are very few studies related to circannual clocks, especially in plants, the evidence indicates that there is still much to be discovered. In the case of *Dendrocalamus strictus* (Rosb.) Nees (Poaceae), seeds were stored at different moisture contents (2.8, 4.7, 6.3, and 8.9%) and storage temperatures (5 °C and −5 °C) in hermetic containers. While no loss in viability was observed, germination was not uniform throughout the three-year study. The seeds showed strong germination during the warm and rainy season (July–August) and decreased germination during the winter months (November–February), possibly regulated by an endogenous rhythm [34].

Another case is reported in dry plants of *Mesembryanthemum nodiflorum* L. (Aizoaceae), with mature seeds enclosed in their dry capsules. These seeds were collected in 1972 in the Sinai Desert, Israel, and stored under dry, room temperature (15 °C–30 °C) and relative humidity (RH) (15–25%). More than 31 years after seed maturation, not embedded, they kept an annual germination rhythm, showing a higher germination percentage in February or March, the appropriate months for germination in the region compared to other months of the year [35]. More recent studies were conducted by Franceschi and collaborators [10] using in vitro germination of four Brazilian orchid species (*Grandiphyllum divaricatum*, *Gomesa praetexta*, *G. forbesii*, and *G. recurva*). Seeds were stored at 4–6% RH and conserved at −20 °C and −80 °C. The authors assessed in vitro germination at four time points: 0, 1, 6, and 12 months. The coloration of the high percentage of embryos in seeds through the triphenyl tetrazolium chloride (TTC) test, related to the low percentage of germination, was interpreted as secondary dormancy. They found that the treatments induced a transition from primary to secondary dormancy, clearly demonstrating that the highest germination percentages occurred at time zero and after 12 months, while lower germination percentages were recorded during intermediate evaluations.

Therefore, the aim of the present study was to analyze the dynamics of seed dormancy in *Laelia speciosa* by in vitro germination and viability testing (using the TTC test), through several evaluations in annual cycles, in previously dehydrated seeds (4% relative humidity) stored at five different temperatures (25 °C, 6 °C, −20 °C, −80 °C, and −196 °C).

## 2. Results

### 2.1. Evaluation of Seed Germination of Laelia speciosa Stored for 36 Months at 25 °C, 6 °C, −20 °C, and −80 °C

At the start of germination (time 0), a common initial germination rate exceeding 95% was recorded for the four temperature conditions (25 °C, 6 °C, −20 °C, and −80 °C). In annual periods (12 and 24 months), Tukey’s mean difference analysis indicated no significant differences, with germination rates ranging from 87% to 94% (Figure 1).

Regarding the percentage values for intermediate months (2, 8, and 18 months) within the first two annual periods, there was a significant decline compared to the annual periods 0, 12, and 24 months (Figure 1a). At 18 months, seeds stored at 25 °C showed a significant decrease in germination, dropping to 36%, with germination continuing to decline in subsequent months, reaching less than 1% after 36 months.

In contrast, seeds stored at 6 °C, −20 °C, and −80 °C (Figure 1b–d) maintained high germination rates at the 24-month cycle. However, during the final evaluation (36 months), only seeds stored at 6 °C retained a germination rate of 85.5%, while those stored at −20 °C (51%) and −80 °C (59%) showed a pronounced decline (Figure 1c,d). In particluar, seeds stored at 25 °C began to experience a pronounced drop in germination as early as 12 months (Figure 1a).

### 2.2. Evaluation of Seed Germination from Six L. speciosa Fruits Stored for 12 Months (0, 3, 6, 9, and 12 Months) at 25 °C and −196 °C

Germination data shown in Figure 2, corresponding to seeds from the second experiment, were analyzed individually for each of the six fruits studied at two storage temperatures (25 °C and −196 °C). Consistent with the behavior observed in Figure 1, the results clearly indicate a circannual pattern under both storage temperatures, with the six-month storage period marking the most significant decline in germination. The extreme storage temperature exhibited a similar pattern to that observed in Experiment 1 for seeds stored at 25 °C and among the germination of the seeds of the six fruits. The extreme freezing storage temperature (-196 °C) recorded a circannual behavior in the seeds of the 6 fruits evaluated, a behavior similar to the same seeds and fruits analyzed in parallel at the 25 °C storage temperature. Likewise, the circannual clock behavior was similar to the response of the first year recorded in experiment 1 at the four temperatures (25 °C, 6 °C, -20 °C, -80 °C) evaluated (Figure 1).

### 2.3. Viability and Germination Tests of Seeds from Six Laelia speciosa Fruits Stored for 12 Months (0, 3, 6, 9, and 12 Months) at 25 °C and −196 °C

The TTC test applied to seeds stored at 25 °C and −196 °C showed viability percentages above 90% for all periods (Figure 3 and Figure 4), with no significant difference observed between periods (*p* > 0.05; Figure 3 and Figure 4). However, this high viability was not reflected in the in vitro germination percentages, which showed significant differences (*p* ≤ 0.01; Figure 2). A marked decline in germination percentages was observed during the intermediate evaluations of the annual period for both storage temperatures (Figure 2, Figure 3 and Figure 4). This discrepancy highlights the presence of dormancy when comparing the results of the two evaluations.

## 3. Discussion

The results observed in the asymbiotic germination response of *Laelia speciosa* seeds revealed a circannual pattern characterized by the difference in response between the red staining of the embryos (>92%) with the TTC in all evaluations and the variable response of the germination percentage, with the germination percentages being reduced during the annual intermediate evaluations (Figure 1 and Figure 2). It is necessary to highlight that the above was recorded under strict conditions both of the relative humidity (4% RH) of the seeds during their storage and of each of the conservation temperatures (25 °C, 6 °C, −20 °C, −80 °C, and −196 °C), without the influence of external physicochemical factors that could affect the physiology of the seeds. Asymbiotic germination showed a circannual rhythm, recording high germination percentages at the beginning and in annual cycles (12 and 24 months), related to the period of the initial establishment of rains in its habitat, and a decrease in germination in the evaluations of the intermediate months, related to the period of stress in the natural environment.

Environmental conditions are fundamental to plant reproduction, serving as key factors regulating seed dormancy and germination [34,35]. Most reports attribute circannual cycles to environmental changes (e.g., humidity, temperature, and light). These factors impact in situ seed banks stored in the soil, with temperature, light, water potential, and nitrate availability being particularly influential [34]. Similarly, seeds stored in laboratory conditions—without dehydration or refrigeration and in containers allowing humidity exchange—also demonstrate circannual rhythms. For example, seeds of *Mesembryanthemum nodiflorum* L. (Aizoaceae) stored dry under room conditions (15–30 °C, 15–25% RH) for 31 years maintained an annual germination rhythm, particularly during February and March, aligning with optimal germination conditions in their native region [35].

However, environmental fluctuations may not be necessary to induce a circannual rhythm. Studies show that this behavior persists in seeds stored under low, stable humidity and temperature conditions. For instance, *Dendrocalamus strictus* seeds stored with varying moisture content (2.8%, 4.7%, 6.3%, and 8.9%) at 5 °C and −5 °C in hermetic containers kept viability but exhibited non-uniform germination over three years. Germination was high during the warm, rainy months (July–August) suitable for the natural site of this species and declined during the winter months (November–February), reflecting a circannual clock [34].

A similar phenomenon was observed in the seeds of some Brazilian orchid species of the genera *Grandiphyllum* and *Gomesa* stored at −20 °C and −80 °C under low humidity conditions [10].

In the first experiment of this study, seeds stored at 6 °C had the highest germination percentage after 36 months compared to the other three storage temperatures. However, a critical question remains: Did the storage temperatures of -20 °C and -80 °C at 36 months induce dormancy or were the seeds already losing viability? Unfortunately, germination percentages were not accompanied by TTC viability assessments, leaving this issue unresolved.

Conventional freezing temperatures (−5 °C to −80 °C), after a few years of storage, have been reported to significantly reduce seed germination percentages in the genus *Cattleya* (Orchidaceae), particularly during storage at −20 °C [36]. For seeds stored at 25 °C, the viability decline after 12 months may be related to the average annual temperature of 18 °C [1] in wild *L. speciosa* populations. Orchid seed conservation has shown success over a decade when seeds are stored at 6 °C [11,12].

According to Harrington’s rule [37], for every 5 °C decrease or increase in storage temperature, seed lifespan doubles or halves, respectively. This suggests that after 12 months, seeds stored at 25 °C would exhibit a decline in viability, consistent with the findings for *L. speciosa* seeds in this study.

In *Disa uniflora* P.J. Bergius seeds stored at 45 °C, viability decreased dramatically from 90% to just 40% within 10 weeks [38].

Seed dormancy, observed as part of a circannual clock, is primarily attributed to genetic mechanisms that respond to annual environmental changes. In our case, it may represent a genetic expression as an adaptive response to evolutionary processes. While significant progress has been made, much remains unclear about the circannual behavior of these species. This aligns with the question posed by Gutterman and Gendler in 2005 [35], which, nearly 20 years later, remains an open inquiry: *How do seeds measure time, and what mechanisms are involved?*

Orchid seeds have been classified as endogenous morphological dormancy [24]. However, *L. speciosa* seeds expressed viability with reddish staining with TTC of the embryos throughout an annual cycle, accompanied by a decrease in the germination percentage in the intermediate evaluations. Strict storage control was maintained at low humidity (4% RH) and five stable temperatures (25 °C, 6 °C, −20 °C, −80 °C, and −196 °C), without the influence of external environmental factors (humidity, light, temperature changes, etc.) during the time they remained stored. The above makes us reflect on the results obtained: Is a circannual latency effect being expressed? To clarify the above, in-depth studies will be needed to determine which factors (genetic, physiological, etc.) intervene in this response of the biological clock, with circannual cycles.

## 4. Materials and Methods

### 4.1. Biological Materials

Seeds of *Laelia speciosa* (Kunth) Schltr. (Orchidaceae) were collected from wild populations in Michoacán, Mexico (Morelia-Quiroga road) at the beginning of fruit dehiscence during two separate periods.

### 4.2. Seed Cleaning and Storage

In both collections, seeds were manually extracted from each fruit and processed individually. They were placed in a desiccator containing silica gel (desiccant) for 3–4 weeks and subjected to repeated trials to achieve the desired moisture content. A relative humidity of 4% was achieved following ISTA testing. The seeds were weighed using an analytical balance (Mettler Toledo Mod. AB 204-S) both before and after drying in an oven (Shel Lab, Cornelius, OR, USA) at 105 °C for 24 h. RH was calculated based on the weight difference [39].

The first collection corresponded to the first experiment. It consisted of seeds from three fruits, which were combined after reaching 4% RH. The seeds were then divided into 32 groups, each placed in a paper envelope and stored in hermetically sealed containers with desiccant at the bottom. Each container was labeled with its designated storage temperature (25 °C, 6 °C, −20 °C, and −80 °C) and the corresponding evaluation time (0, 2, 8, 12, 18, 24, 30, and 36 months). The containers were only removed from storage after the storage time had been completed.

The experimental design for the first experiment was completely randomized, with four treatments corresponding to different storage temperatures (25 °C, 6 °C, −20 °C, and −80 °C). The response variable was the germination percentage at various storage durations (0, 2, 8, 12, 18, 24, 30, and 36 months).

For the second experiment, six fruits were collected and processed individually, following a dehydration method similar to that of the first experiment. Seeds from each fruit were divided in half to evaluate their response to two storage temperatures (25 °C and −196 °C). Each fruit represented an experiment for both temperatures. Each treatment was further divided into 10 portions to monitor seed behavior at five storage periods (0, 3, 6, 9, and 12 months), completing a one-year period. Each evaluation consisted of five replications for germination percentage and five for viability percentage, assessed using TTC (2,3,5-triphenyltetrazolium chloride). Seeds stored at 25 °C were placed in containers similar to those used in the first experiment. For storage at −196 °C, a 60 L cryogenic tank and 1.5 mL Eppendorf tubes were used, with approximately 1500 seeds allocated for 5–6 replications per storage period (0, 3, 6, 9, and 12 months). At the 0-month time point, seeds stored at −196 °C were immersed in liquid nitrogen (LN2) for 24 h. Immersion in LN2 was performed rapidly without the addition of any cryopreservation agents. The Eppendorf tubes were suspended in LN2 using a string labeled at one end with the fruit number and evaluation time, ensuring the seeds were not exposed to temperature fluctuations.

They were stored in different equipment that each recorded a different temperature (25 °C, −196 °C), each of them at 4% RH, subsequently, at each time (0, 3, 6, 9 and 12 months), the sample corresponding to each temperature was taken. All treatments for asymbiotic seed germination were incubated at the same time for each temperature, for both experiments, in the growth chamber (photoperiod of 25 ± 1, 16 h of light (23 μmoles m^−2^ s^−1^) and 8 h of darkness), under a completely random design, taking into account that there may be an edge effect and incidence of light on the incubation surface.

The basal medium used was KC-E, a modified version of the KC medium [40], enriched with micronutrients (in mg L^−1^, 0.056 H_3_BO_3_ (Sigma-Aldrich, St. Louis, MO, USA), 0.016 MoO_3_ (Sigma-Aldrich, St. Louis, MO, USA), 0.040 CuSO_4_ (Sigma-Aldrich, St. Louis, MO, USA), 0.331 ZnSO_4_·7H_2_O (Sigma-Aldrich, St. Louis, MO, USA), 0.083 KI (Sigma-Aldrich, St. Louis, MO, USA), and 0.0025 CoCl_2_·6H_2_O) (Sigma-Aldrich, St. Louis, MO, USA), vitamins from B5 medium [41], 100 mg·L^−1^ of myo-inositol (Sigma-Aldrich, St. Louis, MO, USA), 20 g·L^−1^ of sucrose Grade II (Sigma-Aldrich, St. Louis, MO, USA), and 8 g·L^−1^ of agar (Sigma-Aldrich, St. Louis, MO, USA, A-1296). The pH of the medium was adjusted to 5.0 and sterilized in an autoclave at a pressure of 1.5 kg cm^−2^ for 15 min [42].

### 4.3. Seed Disinfection and Sowing

After the storage treatment, *L. speciosa* seeds were disinfected by placing them inside a filter paper envelope, which was immersed and shaken in 200 mL of 3.5% calcium hypochlorite solution with two drops of Tween 20 for 20 min. Subsequently, under a laminar flow hood, the seeds were rinsed three times with sterile distilled water. The seeds were sown in sterile Petri dishes (15 × 100 mm) containing 30 mL of KC-E culture medium [42]. For both experiments, groups of approximately 200 seeds were used per Petri dish, with five replications for each treatment. The Petri dishes were sealed with Parafilm or sterile plastic tape, labeled with the corresponding temperature and evaluation date, and incubated at 25 ± 1 °C under a 16 h light (23 μmoles·m^−2^·s^−1^) and 8 h dark photoperiod.

### 4.4. Evaluation of Asymbiotic Germination

Approximately 20 days after germination began, the Petri dishes containing germinating seeds were examined. A fixed grid of graph paper was placed at the base of each dish to allow observations in each square centimeter. Germinated and non-germinated seeds were recorded, excluding empty seeds—those visibly lacking an embryo under the microscope. This process was repeated for an average of five replications per treatment. Germination was considered when the embryo was green or had increased in size and had detached from the coat. Non-germinated seeds showed no morphological changes, and the embryo was visible inside the seed.

### 4.5. Evaluation of Seed Viability Using TTC Staining

In the second experiment, seeds stored at 25 °C and −196 °C were tested for viability using the TTC staining method alongside those evaluated for asymbiotic germination. This analysis was carried out for each of the six fruits and at all evaluation times (months) (T0, T3, T6, T9, and T12).

A 0.1% TTC solution with two drops of Tween 20 was added to each seed sample contained in the designated Eppendorf tubes. The seeds were incubated in the solution for 48 h at 25 °C in the dark. Following incubation, five subsamples of approximately 100–200 seeds were taken from each fruit and placed in separate Petri dishes. A millimeter grid was fixed to the bottom of the dishes to facilitate counting of stained (viable) and unstained (non-viable) seeds. Observations were made using a stereoscopic microscope (Zeiss, Oberkochen, Germany: Stemi 2000-C) with a 2× objective. The overall average viability for both storage temperatures was analyzed and compared.

### 4.6. Statistical Analysis

Germination percentage values from the study conducted over 36 months (0, 2, 8, 12, 18, 24, 30, 36) at 25 °C, 6 °C, −20 °C, and −80 °C, as well as from the 12-months (0, 3, 6, 9, 12) study at 25 °C and −196 °C, were transformed using the arcsine transformation [43]. The transformed data were analyzed using one-way ANOVA, followed by Tukey’s test at a 0.05 significance level. Non-transformed average values were used to generate graphical representations of germination behavior over the studied time periods.

The results of the test, which determined the viability of the seeds stained with TTC, were analyzed only in terms of storage time. This is because the responses between temperatures were similar (>98% Viability). The values of both temperatures (25 °C and −196 °C) were joined to search differences among times (0, 3, 6, 9, 12 months) vs. percentage staining.

## 5. Conclusions

*Laelia speciosa* seeds showed a circannual rhythm of germination under controlled laboratory conditions at five storage temperatures (25 °C, 6 °C, −20 °C, −80 °C, and −196 °C) and low relative humidity (4%). This rhythm was characterized by high germination percentages at the beginning and during annual peaks, with declines observed during intermediate evaluations in the annual cycle.

In seeds of *L. speciosa*, the storage temperature of 25 °C was sufficient to cover an annual cycle or period. However, a continuous decline in viability was observed thereafter, with a loss of over 99% recorded at the 36-month evaluation.

At 6 °C, seed viability was best preserved over three years compared to storage at 25 °C, −20 °C, and −80 °C. In contrast, seeds stored at −20 and −80 °C showed a clear trend of viability loss after three years.

Seeds immersed in LN2 show similar behavior in all six fruits, comparable to the results observed at 25 °C for each fruit.

This study represents the first report of a circannual rhythm in the asymbiotic germination of *L. speciosa* seeds under controlled storage conditions at five different temperatures. It is necessary to investigate which factors intervene in the circannual expression of germination physiology.

## Figures and Tables

**Figure 1 plants-14-00336-f001:**
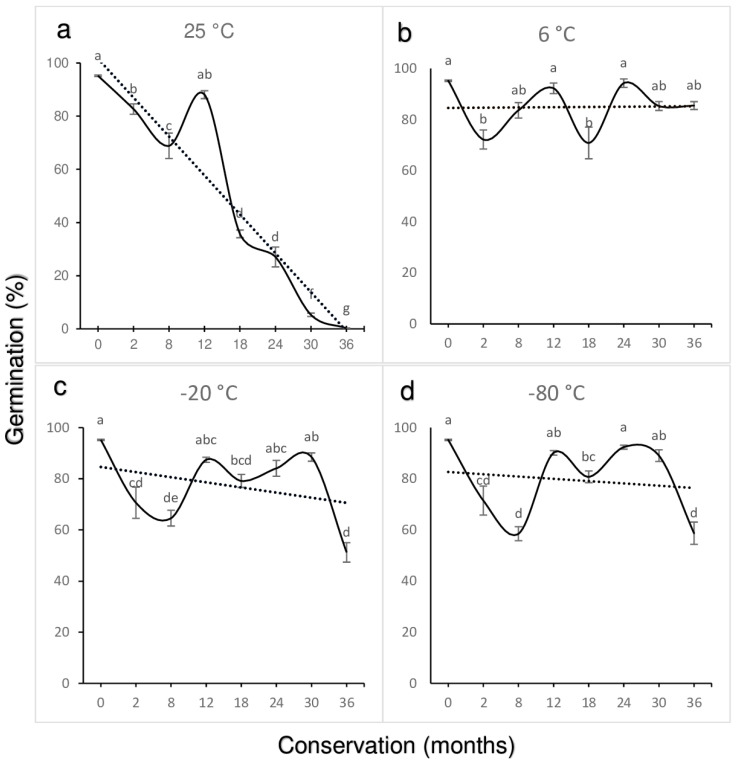
Dormancy behavior was assessed by in vitro germination of *Laelia speciosa* seeds stored for three years and at four storage temperatures ((**a**) = 25 °C, (**b**) = 6 °C, (**c**) = −20 °C, and (**d**) = −80 °C). Bar values with the same letter are not significantly different (*p* ≤ 0.05; Tukey’s test). ANOVA with differences (*p* ≤ 0.01).

**Figure 2 plants-14-00336-f002:**
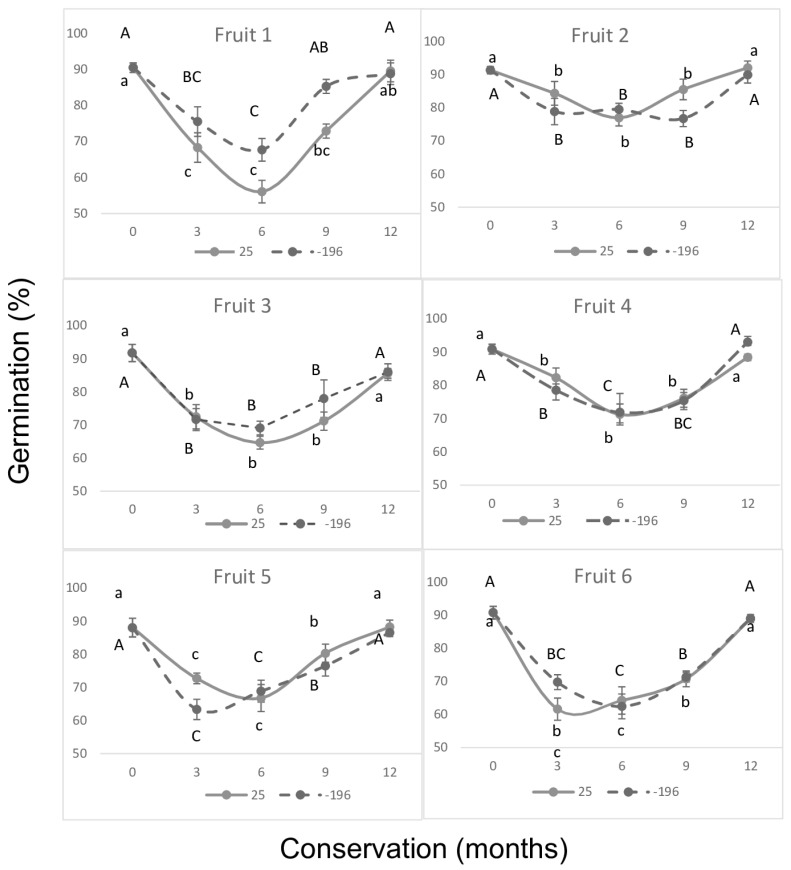
Dormancy behavior was assessed by in vitro germination of *Laelia speciosa* seeds from six fruits of different plants collected in their natural environment and stored at 25 and −196 °C for 12 months. Capital letters correspond to the means from storage at −196 °C and lowercase letters to the means from storage at 25 °C. The bar on the means is equivalent to the standard error. Different letters indicate statistical differences according to the Tukey test (*p* ≤ 0.05). Significant ANOVA (*p* ≤ 0.01).

**Figure 3 plants-14-00336-f003:**
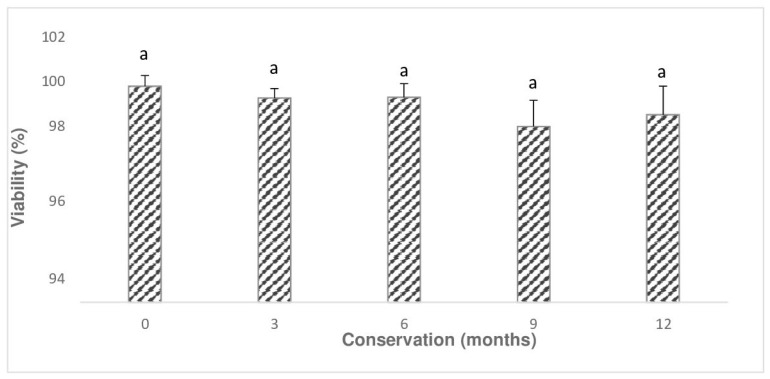
Average viability ± SD in *Laelia speciosa* seeds evaluated by the TTC method in storage at 0, 3, 6, 9, and 12 months. Equal letters without significant difference with 5% reliability by Tukey analysis. SD = standard deviation. ANOVA not significant (*p* > 0.05).

**Figure 4 plants-14-00336-f004:**
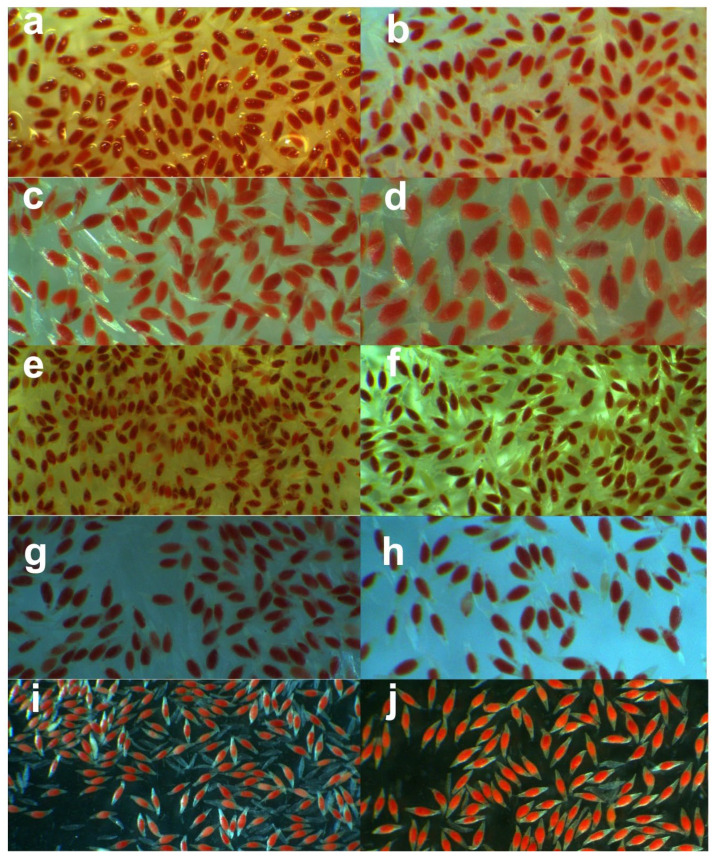
Viability with TTC in *Laelia speciosa* seeds. A mixture of seeds from six fruits was used and stored at 25 and −196 °C. (**a**,**c**,**e**,**g**,**i**): seeds stored at 25 °C for 0, 3, 6, 9, and 12 months, respectively. (**b**,**d**,**f**,**h**,**j**): seeds stored at −196 °C for 0, 3, 6, 9, and 12 months, respectively. No differences were observed between the evaluations (*p* > 0.05).

## Data Availability

The data presented in this study are available on request from the corresponding author.

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
