# Peer review of "Circannual Clock in Laelia speciosa (Orchidaceae) Through Dormancy vs. Germination Dynamics of Seeds Stored Under Controlled Conditions"

_plants, 2025, doi:10.3390/plants14030336_

Round 1

Reviewer 1 Report

Comments and Suggestions for Authors

Comments and Suggestions for Authors

 This study aimed to determine the dynamics of dormancy using triphenyl tetrazolium chloride (TTC) and asymbiotic germination in Laelia speciosa seeds stored for three years and one year at different temperatures. A novel circannual dormancy is proposed. This is the first report of a circannual rhythm in L. speciosa seeds under controlled storage conditions.

The manuscript is constructed according to requirements of “Plants”. The applied research methods are suitable for achieving the research objective. The results are clearly presented and illustrated. Statistical analysis of data obtained additionally support the results. The conclusions made are based on the results.

The following remarks and suggestions can be made:

Abstract

-          I suggest some changes in the part of lines 22-26:”Two experiment were carried out: a) with seeds from wild populations of Laelia speciosa collected at two different times and dehydrated to 4 % relative humidity (RH) and stored for three years at 25 °C, 6 °C, -20 °C, and -80 °C, and b) with seeds from six fruits stored in liquid nitrogen (LN2; 25 -196 °C) at 25 °C for 12 months.”

Introduction

-          I suggests to beggin this part with the sentence in line 41:  Laelia speciosa (H. B. Kunth) … their blooming period [5].” continuing with the sentence in line 38-40: “The best conservation strategy… are a viable alternative [3].”, and omitting the first sentence /lines 36-38/: “With approximately … or medicine [1].”

-          In the sentence of line 41-42”Laelia speciosa (H. B. Kunth) Schltr. (Orchidaceae), regionally known as "flor de mayo" or "flor de Corpus Christi" due to Catholic religious festivities, it is endemic to Mexico”, “it” . is unnecessary. And, in the next sentence, Laelia speciose to be replaced with “it”

-          After the part that end in “…can survive on the planet [20].” /line 70/ it is necessary to include information about orchid seed dormancy, which will serve as a transition to the next part related to seed dormancy.

-          The part in lines 101-116 :”In birds, … survival [34,35]..” to be omitted and continue with the sentence of line 117.

-          The sentence in lines 135-137:”Viability, determined by the triphenyl tetrazolium chloride (TTC) test, was interpreted as dormancy, considering the difference between germination and viability.” needs clarification: viability cannot be interpreted as dormancy, since /as described above in the text/ dormant seeds are seeds that do not have the ability to germinate within a certain period of time, and viability refers to the respiratory activity of the seed embryo, i.e. whether the embryo is alive or not; thus dormant seeds are alive.

-          My suggestion for the last part of this section /lines 141-144/ is the following: “Therefore, the aim of the present study was to analyze the dynamics of seed dormancy of Laelia speciosa by testing asymbiotic germination and viability (using TTC test), through several assessments in annual cycles, in seeds previously dehydrated (4% relative humidity) and stored at five different temperatures (25 °C, 6 °C, -20 °C, -80 °C and -196 °C).”        

Materials and Methods

-          The part in lines 146-149: Biological Materials. to be redacted as follows: Seeds of Laelia speciosa (Kunth) Schltr. (Orchidaceae) were collected from wild populations in Michoacán, Mexico (Morelia-Quiroga road) at the beginning of fruit dehiscence during two separate periods.

-          The explanation “those visibly lacking an embryo under the microscope” in line 206 is unnecessary to explain what an empty seed is

-          How was the starting number of fruits chosen for the experiments: why were three fruits taken for the first experiment and six for the second? Wouldn't it be better to take a certain number of seeds and distribute them among the individual experiments?

In conclusion, this manuscript is recommended for publication in “Plants”.

Author Response

REVIEWER 1

 Reviewer 1. - I suggest some changes in the part of lines 22-26: “Two experiments were carried out: a) with seeds from wild populations of Laelia speciosa collected at two different times and dehydrated at 4% relative humidity (RH) and stored for three years at 25 °C, 6 °C, -20 °C and -80 °C, and b) with seeds from six fruits stored in liquid nitrogen (LN2; 25 -196 °C) at 25 °C for 12 months.”

Author: Changes were made in the order and wording proposed by the reviewer in the abstract, which was as follows: Highlights version: lines 27-34; clean version: lines 20-27.

Reviewer's proposal: This study aimed to determine the dynamics of dormancy using triphenyl tetrazolium chloride (TTC) and asymbiotic germination in Laelia speciosa seeds stored for three years and one year at different temperatures. A novel circannual dormancy is proposed. This is the first report of a circannual rhythm in L. speciosa seeds under controlled storage conditions. Two experiments were carried out: a) with seeds from wild populations of Laelia speciosa collected at two different times and dehydrated to 4 % relative humidity (RH) and stored for three years at 25 °C, 6 °C, -20 °C, and -80 °C, and b) with seeds from six fruits stored in liquid nitrogen (LN2; -196 °C) at 25 °C for 12 months. Germination conditions: 25±1 °C with 16 hours of light (23 µmol m-2 s-1), 8 hours of darkness for both trials. Circannual dormancy was present in viable seeds (>92 %) throughout an annual cycle and with a significant decrease in the percentage of asymbiotic germination in evaluations distant from the beginning of the rainy season in the natural population.

(The Abstract was completed with the previous sentence highlighted in yellow. Clean version: Lines: 28-30; Highlights version: lines: 58-61.

Author: The reviewer's input was considered in its entirety.

Reviewer 1: Line 41. - I suggest starting this part with the sentence in line 41: “Laelia speciosa (H. B. Kunth) … its flowering period [5]”. continuing with the sentence in lines 38-40: “The best conservation strategy… are a viable alternative [3]”, and omitting the first sentence /lines 36-38/: “With approximately… or medicine [1]”.

- In the sentence on line 41-42 “Laelia speciosa (H. B. Kunth) Schltr. (Orchidaceae), known regionally as “May flower” or “Corpus Christi flower” due to Catholic religious holidays, is endemic to Mexico”, “it” is unnecessary.

Author: All proposed changes were made and incorporated in the text in the order and changes suggested by reviewer 1Line 43: Replace the name of Laelia speciosa with it. clean version: Lines 35-48; Highlights version: Lines 47-71.

Reviewer 2. requested that information on orchid seed dormancy be added as an introduction to what follows on dormancy and the circadian clock. Clean version: Line 66, 67; Highlights version:

Reviewer 1: Line 70: After the part that ends in “…can survive on the planet [20].”/line 70/ it is necessary to include information about orchid seed dormancy, which will serve as a transition to the next part related to seed dormancy.

Author: Information about the type of seed latency was included on Clean version: lines 64,65; and Highlights version: Lines 76, 77.

Reviewer 1:- The part on lines 101-116: “In birds, … survival [34,35].” should be omitted and continued with the sentence on line 117.

Author: That entire text was omitted and continued with Highlights version: line 111-127.

Reviewer 1: - The sentence on lines 135-137: “Viability, as determined by the triphenyl tetrazolium chloride (TTC) test, was interpreted as dormancy, considering the difference between germination and viability.” needs clarification: viability cannot be interpreted as dormancy, since /as described above in the text/ dormant seeds are seeds that do not have the ability to germinate within a given period of time, and viability refers to the respiratory activity of the seed embryo, i.e. whether the embryo is alive or not; Therefore, dormant seeds are alive.

Author: The viability evaluation was carried out only with healthy seeds, with embryos, and with the same moisture content and considering that Baskin and Baskin (2004), have indicated that seeds that do not germinate, even when environmental conditions are favorable, are considered dormant, the assessment that a dormant seed behavior is observed, as a result of the difference between germination and viability percentages evaluated, is valid and reliable.

Reviewer 1:- My suggestion for the last part of this section /lines 141-144/ is the following: “Therefore, the objective of the present study was to analyze the dynamics of the dormancy of Laelia speciosa seeds by means of germination and asymbiotic viability tests (using the TTC test), through several evaluations in annual cycles, in previously dehydrated seeds (4% relative humidity) and stored at five different temperatures (25 °C, 6 °C, -20 °C, -80 °C and -196 °C)”.

Author: The suggestion was considered and included in the text. Highlights version: Lines 161-165. clean version: Lines 132-136.

Reviewer 1:

Materials and methods

- The part on lines 146-149:

Biological materials. will be written as follows: Seeds of Laelia speciosa (Kunth) Schltr. (Orchidaceae) were collected from wild populations in Michoacán, Mexico (Morelia-Quiroga highway) at the beginning of fruit dehiscence during two separate periods.

Author: Lines 146-149 reviewer is suggestion was addressed. Clean version: Lines 279-281; Highlights version: Lines 296-298.

Reviewer 1: - The explanation “those that visibly lack an embryo under the microscope” in line 206 is unnecessary to explain what an empty seed is.

Author: This consideration could not be included, given that there are fruits with a high index of empty seeds, therefore they should not be considered in a germination test; typical of certain individuals or plants, due to different factors that condition this characteristic.

Reviewer 2 Report

Comments and Suggestions for Authors

Overall, this study provides valuable insights into the dormancy dynamics of Laelia speciosa seeds and contributes to our understanding of circannual rhythms in plants. The study addresses an important gap in knowledge regarding circannual dormancy in orchid seeds, particularly for the endangered species Laelia speciosa.

Addressing the suggested comments could further enhance the scientific impact of this research.

Line 20-22: consider editing it as follows: The objective of this research was to investigate how dormancy changes in Laelia speciosa seeds after storage periods of one and three years at various temperatures. The study employed two methods to assess seed viability and dormancy: the triphenyl tetrazolium chloride (TTC) test and asymbiotic germination techniques.

Line 21: Laelia speciosa (Kunth) Schltr. 

Line 46: with fruits...

Line 66-68: This sentence fragment should be connected to the previous sentence. "which appear to be negative, were compensated by evolving in association with mycorrhizal fungal symbionts [19], a strategy that has made them highly successful."

To enhance the readability and scientific soundness of manuscript please consider the following suggestions:

Consider adding a brief explanation of the circannual clock concept earlier in the introduction to provide context for readers unfamiliar with the term.

The methods section could benefit from more detailed descriptions of the germination conditions and TTC analysis procedures.

Include a brief explanation of why these specific storage temperatures were chosen for the experiments.

Line 234: should be "Seed"

Author Response

REVIEWER 2

Reviewer 2: Lines 20-22: Change target

Author: Modifications were made according to the proposal of reviewer 1, and both considerations are very similar. Clean version:132-136; Highlights version: lines 161-165.

Reviewer 2: Line 21: Laelia speciosa (Kunth) Schltr.

Author: Authors attached to species Reviewer 1: Highlights version: lines 28-29; Clean version: line 21.

Reviewer 2: Line 46: with fruits...

Author: The word fruits were made plural. Highlights version: line 52; Clean version: line 39.

Reviewer 2: Lines 66-68: This sentence fragment should be connected to the previous sentence. "which appear to be negative, were compensated by evolving in association with mycorrhizal fungal symbionts [19], a strategy that has made them highly successful."

Author: Lines 66-68 attended to, changing the period to a comma, which provides continuity. Highlights version: line 75; Clean version: line 62.

Reviewer 2: To improve the readability and scientific soundness of the manuscript, consider the following suggestions:

Consider adding a brief explanation of the concept of the circannual clock at the beginning of the introduction to provide context for readers who are not familiar with the term.

Author: In lines 89-93 in the original version the circannual clock is described, it is expanded, and moving it to the beginning alters the order, we wish to present to the reader the problems of the species under study. Highlights version: lines 99-104; Clean version: lines 85-91.

Reviewer 2: The methods section could benefit from more detailed descriptions of germination conditions and TTC analysis procedures.

Author: The research goes beyond the simple physiology of germination, information that can be obtained in various reports, in which we attach the citation of one of our works where we describe germination and tissue culture (Ramos-Ortiz et al., 2020). We prefer to focus the reader's attention on the subject of circannual rhythm.

Reviewer 2: Include a brief explanation of why these specific storage temperatures were chosen for the experiments.

Author: Three fruits because the conservation of seeds stored at 4 temperatures was going to be evaluated for 3 years and it is a significant number, three is also a significant number to represent the behavior of the species in the population studied. Six fruits for this case we needed to see the individual behavior of a representative number of individuals of the species, which allowed us to see the same response in the 6 fruits and avoid questions due to lack of repetitions. And, these temperatures were chosen because they are the most commonly reported in the storage and conservation of orchid seeds.

Reviewer 2: Line 234: should be Sed

Author: The error from sed to seed was corrected. Clean version Line 138; Highlights version: line 170.
